

# Device-independent and semi-device-independent entanglement certification in broadcast Bell scenarios

Emanuel-Cristian Boghiu[1*], Flavien Hirsch[2*], Pei-Sheng Lin[3],
Marco Túlio Quintino[4,2,5] and Joseph Bowles[1]

**1** ICFO - Institut de Ciencies Fotoniques, The Barcelona Institute of Science and Technology,
08860 Castelldefels (Barcelona), Spain
**2** Institute for Quantum Optics and Quantum Information (IQOQI),
Austrian Academy of Sciences, Boltzmanngasse 3, 1090 Vienna, Austria
**3** Department of Physics and Center for Quantum Frontiers of Research & Technology
(QFort), National Cheng Kung University, Tainan 701, Taiwan
**4** Sorbonne Université, CNRS, LIP6, F-75005 Paris, France
**5** Faculty of Physics, University of Vienna, Boltzmanngasse 5, 1090 Vienna, Austria

★ These authors have contributed equally.

## Abstract

It has recently been shown that by broadcasting the subsystems of a bipartite quantum state, one can activate Bell nonlocality and significantly improve noise tolerance bounds for device-independent entanglement certification. In this work we strengthen these results and explore new aspects of this phenomenon. First, we prove new results related to the activation of Bell nonlocality. We construct Bell inequalities tailored to the broadcast scenario, and show how broadcasting can lead to even stronger notions of Bell nonlocality activation. In particular, we exploit these ideas to show that bipartite states admitting a local hidden-variable model for general measurements can lead to genuine tripartite nonlocal correlations. We then study device-independent entanglement certification in the broadcast scenario, and show through semidefinite programming techniques that device-independent entanglement certification is possible for the two-qubit Werner state in essentially the entire range of entanglement. Finally, we extend the concept of EPR steering to the broadcast scenario, and present novel examples of activation of the two-qubit isotropic state. Our results pave the way for broadcast-based device-independent and semi-device-independent protocols.



## Contents



# 1 Introduction

One of the most fascinating aspects of quantum theory is the fact that it does not obey the same common-sense form of causality that is observed at the macroscopic level. This is made formal in Bell's theorem [1], which proves that any attempt to reformulate the theory in a classical picture of reality is doomed to fail at reproducing the predictions of certain experiments, called Bell tests. In a Bell test, an entangled quantum system is prepared and shared between a number of spatially separated laboratories and subsequently measured. Remarkably, measurements made in these separate locations lead to correlations between outcomes that defy explanation via shared classical resources alone: a phenomenon first shown by Bell [1] and consequently called Bell nonlocality (see [2] for a review article).

The discovery of Bell nonlocality has since developed into its own field of research, and much is now known. This research program has also inspired new notions of non-classicality that are closely related to Bell nonlocality. The most widely studied of these is EPR steering [3–8]. Like Bell nonlocality, EPR steering is a form of non-classicality exhibited by entangled quantum states, and relates to the fact that a measurement made on one subsystem of an entangled state has the ability to influence or "steer" the distant quantum state of another subsystem. EPR steering can also be understood from the perspective of Bell nonlocality, where one makes

stronger assumptions about the physics of one of the devices; for this reason, the phenomenon is generally easier to observe in experiments than Bell nonlocality [9, 10].

Aside from foundational implications, Bell nonlocality and EPR steering also play a key role in quantum information technologies. In particular, the phenomena serve as the fuel for the class of device-independent (DI) [11–18], and certain types of semi-device-independent (SDI) protocols [19–22]. The most basic of these protocols is that of entanglement certification: since both phenomena require the use of entangled states, the observation of either implies a certificate of entanglement in the underlying physics. Device-independent certification of entanglement is highly desirable, since it allows us to ensure entanglement even in a scenario where the measurements performed by the parties are untrusted and uncharacterised. Additionally, such protocols serve as starting points for advanced protocols of cryptography [13, 23, 24], randomness certification [25, 26], and randomness amplification [27, 28], in which their security is not based on the internal mechanism of the devices but on the fact that events from distant parties are space-like separated. An interesting growing body of work is also showing how Bell nonlocality plays a key role in quantum computational advantages [29–32].

Basic questions regarding both Bell nonlocality and EPR steering still remain open however. Perhaps the simplest of these is the one asking which entangled states are capable of exhibiting these forms of non-classicality. In particular, it is known that entanglement alone is not sufficient to observe neither Bell nonlocality nor EPR steering, since some mixed entangled states are known to admit so-called local hidden-variable, or local hidden state models [33–35]. A clearer answer to this question is desirable from a foundational perspective, but also from a technological perspective, given their connection to quantum information technologies.

An important discovery in this respect was that of activation. The basic message is as follows: some quantum states that show only classical behaviour in the orthodox "standard scenario" can have their non-classicality activated, or revealed, by subjecting the state to a more complex measurement scenario. This both expands the set of entangled states that exhibit non-classical behaviour, and rekindles the hope of proving Bell nonlocality or EPR steering of all entangled states. There are a number of different methods that have been shown to activate quantum states (see [36] for a more detailed discussion). In Refs. [37, 38], it is shown that Bell nonlocality can be activated by applying local filters to the state before a Bell test. This can be seen as a specific case of the more general sequential measurement scenario, in which a sequence of time-ordered measurements is made on the local subsystems of the state [39]. Later, it was shown that activation of Bell nonlocal and EPR-steering is also possible by taking multiple copies of the state, and performing joint measurements on the local subsystems [40, 41]. This method appears to be more powerful than the sequential scenario [42], which is perhaps to be expected given the additional resources and entanglement granted by the multiple copies.

Recently, a new technique based on broadcasting was discovered and shown to lead to Bell nonlocality activation [36] (see also [43] for a prior related work which inspired the definition of broadcast nonlocality). In this scenario, one or more of the local subsystems is broadcast to a number of additional parties (see Fig. 1). The entanglement present in the original state is thus shared between a larger number of parties, and interestingly, this can be used to activate the Bell non-locality of the original state. The broadcast scenario also appears to be significantly more powerful than the sequential measurement scenario: for instance, for the two-qubit Werner state, broadcasting leads to activation of Bell nonlocality for significantly lower visibilities [36]. This has practical implications, since although stronger examples of activation are known by using many copies, the broadcast scenario requires the manipulation of a single copy of the state per experimental round, does not require joint measurements, and may thus admit a simpler implementation.

In this article we build on this initial work, and prove a number of new results related to nonlocality, device-independent (DI) and semi-DI entanglement certification, which we

summarize here.

- **Bell nonlocality in broadcast scenarios**—We give two methods to construct Bell inequalities tailored to the broadcast Bell scenario, starting from a Bell inequality in the standard scenario. We also study detector inefficiencies in the broadcast scenario. For the case of the two-qubit maximally entangled state $|\Phi^+\rangle = [|00\rangle + |11\rangle]/\sqrt{2}$, we show how one can demonstrate Bell nonlocality with lower detection inefficiencies than in the standard scenario.

- **Stronger activation through broadcasting**—We prove a stronger notion of activation than previously shown in [36]. More precisely, we show that through broadcasting, it is possible to convert a state with a local hidden-variable (LHV) model for general (POVM) measurements, to a state this exhibits genuinely multipartite nonlocal correlations. This is probably the most extreme "jump" in Bell nonlocality class that has been demonstrated using a single quantum state. Such a result highlights the extent to which notions of locality in the standard scenario (i.e. the existence of an LHV model) are unable to capture the strongly nonlocal properties of entangled states.

- **Device-independent entanglement certification**—We investigate device-independent entanglement certification in the broadcast scenario. In Ref. [36] it was shown that broadcasting allows entanglement certification for noise thresholds much lower than previously known. For the case of the isotropic state of two qubits (local unitary equivalent to the two-qubit Werner state [33]),

$$\rho(\alpha) = \alpha|\Phi^+\rangle\langle\Phi^+| + (1-\alpha)\mathbb{1}/4, \tag{1}$$

it was shown that DI entanglement certification is possible for visibilities $\alpha > \frac{1}{2}$. Here, we show that this can in fact be extended to visibilities greater than $\alpha > 0.338$, using a numerical technique based on the NPA hierarchy. Since the state is entangled for $\alpha > \frac{1}{3}$, this is essentially the entire range of entanglement, and we suspect that this could be lowered arbitrarily close to $\alpha = \frac{1}{3}$ with more computational power. This suggests the possibility of designing device-independent protocols with much greater tolerance to noise, which is highly desirable given the high experimental requirements that hinder device-independent protocols.

- **Broadcast steering**—We extend the definition of EPR steering to the broadcast scenario, and study the phenomenon using the two-qubit isotropic state, showing that broadcast steering is possible for visibilities greater than 0.4945 when broadcasting to two parties and visibility greater than 0.4679 when broadcasting to three parties. This is below the threshold of $\frac{1}{2}$ below which the state has a local hidden state model in the standard scenario [3,33], and is the first example of the activation of steering using a single copy of this state.

## 2 The broadcast scenario

Here we describe Bell nonlocality in the broadcasting scenario, starting with the definition of standard bipartite Bell nonlocality.

### 2.1 Bell nonlocality in the standard scenario

In a Bell scenario, Alice and Bob are distant parties that can perform various local measurements on a shared physical system. We denote by $x$ the choice of measurements performed by Alice

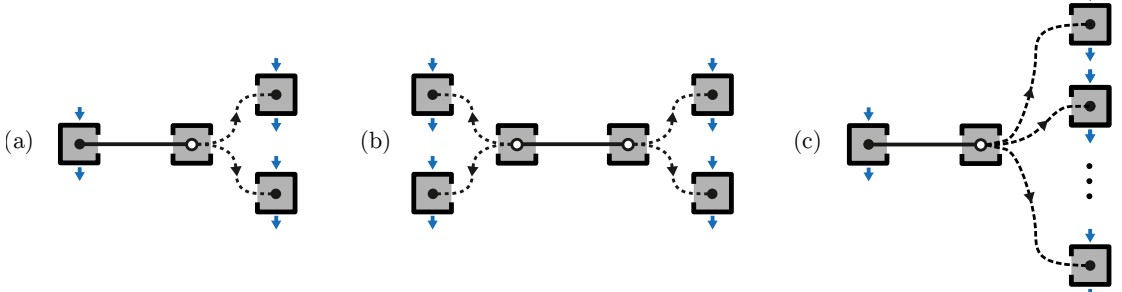

Figure 1: Three broadcast scenarios. One (or more) of the local systems is broadcast via the application of a quantum channel, resulting in a multipartite state, sent to distant parties. Local measurements are then performed on this state, and the resulting statistics are used to rule out a local hidden-variable description for the original bipartite state.

and $a$ the output received by Alice by performing her measurement. Analogously, $y$ and $b$ stand for the choice of respective measurement and outcome performed by Bob. The probability of Alice and Bob obtaining the outcomes $a$ and $b$ after performing the measurements $x$ and $y$ is described by $p(ab|xy)$ and the set of all probabilities on a given scenario is referred to as a behaviour $\{p(ab|xy)\}$.

A behaviour with probabilities $p(ab|xy)$ is Bell local if it can be explained by a classical mixture of independent strategies which only depend on their local input. More formally, if the probabilities can be explained by the following: Alice and Bob share a physical system, sometimes referred to as a hidden variable, which assumes the value $\lambda$ with probability density $\Pi(\lambda)$. Whenever Alice performs the measurement labelled by $x$, she outputs $a$ with probability $p_A(a|x, \lambda)$. Whenever Bob performs the measurement labelled by $y$, he outputs $b$ with probability $p_B(b|y, \lambda)$. In this way, the behaviour $\{p(ab|xy)\}$ admits a Bell local decomposition if it can be written as

$$p(ab|xy) = \int \Pi(\lambda)\, p_A(a|x, \lambda)\, p_B(b|y, \lambda)\, d\lambda. \tag{2}$$

In a seminal work [1], Bell showed that quantum correlations do not necessarily respect (2), a phenomenon now known as Bell nonlocality. More formally, let $\rho_{AB}$ be a quantum state shared by Alice and Bob, *i.e.*, $\rho_{AB}$ is a positive semidefinite operator, $\rho_{AB} \geq 0$, with unit trace, $\mathrm{Tr}(\rho_{AB}) = 1$. Let $\{A_{a|x}\}$ be a set of POVMs representing the quantum measurement $x$ with output $a$, *i.e.* $A_{a|x}$ is a positive semidefinite operator, $A_{a|x} \geq 0$, and the set $\{A_{a|x}\}_a$ respects the normalization constraint $\sum_a A_{a|x} = \mathbb{1}, \forall x$. Analogously, $\{B_{b|y}\}$ is a set of POVMs representing Bob's measurements. The probability of Alice and Bob obtaining the outcomes $a$ and $b$ when performing the quantum measurements on their shared system is given by

$$p(ab|xy) = \mathrm{Tr}(\rho_{AB} A_{a|x} \otimes B_{b|y}). \tag{3}$$

Bell theorem thus states that there exist quantum states and measurements such that this behaviour does not admit a Bell local decomposition as in Eq. (2) [1, 2, 44].

A quantum state $\rho_{AB}$ is separable if it admits a decomposition

$$\rho_{AB} = \int \Pi(\lambda)\, \rho_A^\lambda \otimes \rho_B^\lambda\, d\lambda, \tag{4}$$

where $\rho_A^\lambda$ and $\rho_B^\lambda$ are quantum states and $\Pi$ is a probability density. States which are not separable are denoted as *entangled*. From (2) it is easy to see that separable states can only lead

to Bell local behaviours, hence if a quantum behaviour does not admit a Bell local decomposition, we certify that Alice and Bob share an entangled state.

Interestingly, there exist quantum states which, despite being entangled, can only lead to Bell local correlations in standard Bell scenarios [7, 8, 33, 34, 45–47]. That is, there are entangled states $\rho_{AB}$ such that for every possible choice of local measurements performed by Alice and Bob, the behaviour given by $p(ab|xy) = \text{Tr}(\rho_{AB}A_{a|x} \otimes B_{b|y})$ is necessarily Bell local (i.e. admits a decomposition (2)). Such quantum states are said to admit a local-hidden-variable (LHV) model. Over the years, researchers have proposed extended scenarios that are able to "activate" the Bell nonlocality of quantum states. That is, states which have an LHV model, hence Bell local in standard Bell tests, may display nonlocal correlations in more complex scenarios. Examples of these scenarios include allowing local filtering operations before the Bell test [37, 42], applying sequential measurements [39, 48], using multiple copies of the shared state [40] or distributing the bipartite state in networks [49, 50].

## 2.2 Bell nonlocality in broadcast scenarios

Recently, Ref. [36] proposed a Bell scenario entitled *broadcast nonlocality* which brings novel insights to quantum nonlocality and can also activate Bell nonlocality for some entangled states with an LHV model. We now present the concept of broadcast nonlocality by starting with the scenario illustrated in Fig. 1a). Let $\rho_{AB_0}$ be a quantum state shared between Alice and Bob$_0$ (denoted by the black line in the figure). A quantum channel is performed on the $B_0$ subsystem which will "broadcast" this system to two distant parties, Bob and Charlie. More formally, let $\Omega_{B_0 \to BC}$ be a quantum channel, *i.e.,* a completely positive trace preserving (CPTP) channel, that enlarges the $B_0$ space to a tensor product space of $B$ and $C$. Due to this property of transforming a single system into multiple systems, we refer to this channel as "broadcast channel". Examples of such channels are approximate cloning [51, 52] or broadcasting of quantum information as in Ref. [53], although the channel need not be of this form. When the quantum state $\rho_{AB_0}$ undergoes a broadcast channel $\Omega_{B_0 \to BC}$, the resulting tripartite state is

$$\rho_{ABC} = \mathbb{1} \otimes \Omega_{B_0 \to BC}[\rho_{AB_0}]. \tag{5}$$

Let $A_{a|x}$, $B_{b|y}$, and $C_{c|z}$ be POVMs representing quantum measurements performed by Alice Bob, and Charlie respectively. The probabilities of obtaining outcomes $a, b, c$ when measurements labelled by $x, y, z$ are performed are given by

$$p(abc|xyz) = \text{Tr}(A_{a|x} \otimes B_{b|y} \otimes C_{c|z} \, \rho_{ABC}). \tag{6}$$

In a standard bipartite Bell scenario under the bipartition $A|BC$, we would say that this behaviour is Bell nonlocal if it cannot be written as

$$p(abc|xyz) = \int \Pi(\lambda) \, p_A(a|x, \lambda) \, p_B(bc|yz, \lambda) \, d\lambda. \tag{7}$$

The key difference is that, in a broadcast scenario, we assume that, since Bob and Charlie are far apart, they are restricted by non-signalling constraints. That is, the broadcast channel represented by $\Omega$ may provide Bob and Charlie strong non-signalling correlations (even supra-quantum ones, such as PR-box correlations [54–56]), but they are necessarily non-signalling. This is needed to ensure that the non-classicality in the observed correlations is not a result of the transformation device alone (See Ref. [36] for a more detailed discussion). In mathematical terms, a behaviour $\{p(abc|xyz)\}$ is broadcast nonlocal if it cannot be written as

$$p(abc|xyz) = \int \Pi(\lambda) \, p_A(a|x, \lambda) \, p_{BC}^{\text{NS}}(b, c|y, z, \lambda) \, d\lambda, \tag{8}$$

where the behaviour $\{p_{BC}^{\mathrm{NS}}(b, c|y, z, \lambda)\}$ respect the non-signalling constraints which are given by

$$\sum_b p_{BC}^{\mathrm{NS}}(bc|yz, \lambda) = \sum_b p_{BC}^{\mathrm{NS}}(bc|y'z, \lambda) \; \forall y, y', c, z, \lambda, \tag{9}$$

$$\sum_c p_{BC}^{\mathrm{NS}}(bc|yz, \lambda) = \sum_c p_{BC}^{\mathrm{NS}}(bc|yz', \lambda) \; \forall z, z', b, y, \lambda. \tag{10}$$

In a similar vein, one can also consider a broadcast scenario where both sides (Alice and Bob) perform broadcast channels before the Bell test, as in example b) of Fig. 1, or that the broadcast channel broadcasts the state into multiple parties, as in example c) of Fig. 1. The corresponding definitions of broadcast nonlocality in these scenarios follow the same logic as above, by allowing parties that share a common broadcast channel to share non-signalling resources.

## 3 Novel results and methods for broadcast nonlocality

### 3.1 Promoting standard Bell inequalities to the broadcast scenario

Here we give a method to construct Bell inequalities tailored to the broadcast scenario, starting from Bell inequalities defined in the standard scenario. In the broadcast scenario, it is shown that one can activate nonlocality for the isotropic state, $\rho_\alpha = \alpha \left|\Phi^+\right\rangle\left\langle\Phi^+\right| + (1-\alpha)\mathbb{1}/4$, for $\alpha > \frac{1}{\sqrt{3}}$ [36]. This is certified by the inequality

$$\begin{aligned}
\mathcal{I} = {} & \langle A_0 B_0 C_0 \rangle + \langle A_0 B_1 C_1 \rangle + \langle A_1 B_1 C_1 \rangle - \langle A_1 B_0 C_0 \rangle \\
& + \langle A_0 B_0 C_1 \rangle + \langle A_0 B_1 C_0 \rangle + \langle A_1 B_0 C_1 \rangle - \langle A_1 B_1 C_0 \rangle \\
& + 2\langle A_2 B_1 \rangle - 2\langle A_2 B_0 \rangle \leq 4,
\end{aligned} \tag{11}$$

written in the standard correlator notation, where

$$\langle A_x B_y C_z \rangle = \sum_{a, b, c = \pm 1} abc \, p(abc|xyz), \tag{12}$$

$$\langle A_x B_y \rangle = \sum_{a, b = \pm 1} ab \, p(ab|xy), \tag{13}$$

$$\langle A_x \rangle = \sum_{a = \pm 1} a \, p(a|x). \tag{14}$$

Similar definitions hold for $\langle A_x C_z \rangle$ and $\langle B_y C_z \rangle$, and for $\langle B_y \rangle$ and $\langle C_z \rangle$. The best quantum violation found for this inequality is $4\sqrt{3}$ with the measurements and channel given in Ref. [36] and using the isotropic state with $\alpha = 1$.

Ineq. (11) can be restructured as follows

$$\langle \mathrm{CHSH}[A_0, A_1, C_0, C_1](B_0 + B_1) \rangle + \mathcal{L}_{\mathrm{CHSH}} \langle A_2(B_1 - B_0) \rangle \leq 2\mathcal{L}_{\mathrm{CHSH}}, \tag{15}$$

where

$$\mathrm{CHSH}[A_0, A_1, C_0, C_1] := (A_0 - A_1)C_0 + (A_0 + A_1)C_1, \tag{16}$$

and $\mathcal{L}_{\mathrm{CHSH}}$ denotes the local bound of the CHSH inequality in the standard Bell scenario, *i.e.*, $\mathcal{L}_{\mathrm{CHSH}} = 2$. Here, we slightly abuse notation so that for example $\langle A_2(B_1 - B_0) \rangle$ is understood as $\langle A_2 B_1 \rangle - \langle A_2 B_0 \rangle$. The form of (15) suggests the following recipe for promoting any two-outcome Bell inequality $\mathcal{I}[A_0, \ldots, A_m, C_0, \ldots, C_k]$ to the broadcast scenario through the following *ansatz*

$$\langle \mathcal{I}[A_0, \ldots, A_m, C_0, \ldots, C_k](B_0 + B_1) \rangle + \mathcal{L}_{\mathcal{I}} \langle A_{m+1}(B_1 - B_0) \rangle \leq 2\mathcal{L}_{\mathcal{I}}, \tag{17}$$

where $\mathcal{L}_{\mathcal{I}}$ is the local bound of $\mathcal{I}$ in the standard scenario. We prove in Appendix D that (17) is valid so long as the Bell inequality $\mathcal{I}$ does not contain any 1-body correlator terms $\langle A_x \rangle$ for Alice.

The same procedure can also be applied to the 4-partite symmetric broadcast scenario of Fig. 1b). In [36, Sec. 4.2] an inequality is given for this scenario which can be written

$$\langle \text{CHSH}[A_0, A_1, C_0, C_1](B_0 + B_1)D_0 \rangle + \mathcal{L}_{\text{CHSH}}\langle (B_1 - B_0)D_1 \rangle \leq 2\mathcal{L}_{\text{CHSH}}, \qquad (18)$$

where Alice and Bob are on the left side and Charlie and Dave are on the right side, as in Fig. 1b). This inequality is also violated by the isotropic state for $\alpha > \frac{1}{\sqrt{3}}$. Similarly to above, this suggests the construction

$$\langle \mathcal{I}[A_0, \ldots, A_m, C_0, \ldots, C_k](B_0 + B_1)D_0 \rangle + \mathcal{L}_{\mathcal{I}}\langle (B_1 - B_0)D_1 \rangle \leq 2\mathcal{L}_{\mathcal{I}}. \qquad (19)$$

We prove in Appendix D that (19) is valid so long as the Bell inequality does not contain any 1-body correlator terms $\langle A_x \rangle$ or $\langle C_z \rangle$.

We now apply these constructions to two well-known Bell inequalities in the standard scenario, and study noise resistance with respect to the isotropic state (1). The two Bell inequalities we consider are (i) the chained Bell inequality [57,58] in the case of both parties having 3 input settings, and (ii) the elegant Bell inequality [59], where Alice has 3 inputs and Bob 4 inputs. These inequalities are both maximally violated by the maximally entangled state ($\alpha = 1$), and are violated by the isotropic state for (i) $\alpha > \frac{4}{6\cos\frac{\pi}{6}} \approx 0.7698$, and (ii) $\alpha > \frac{6}{4\sqrt{3}} \approx 0.8660$ respectively. Via a numerical see-saw optimization in the broadcast scenario, we have found that the corresponding inequalities (17) are violated in the broadcast scenario for visibilities (i) $\alpha > 0.6100$ and (ii) $\alpha > 0.6799$. Surprisingly, the same bounds are obtained using the construction (19). Notice that for both examples this visibility is below $\alpha = 1/K_3$ where $0.683 < 1/K_3 < 0.697$ and $K_3$ is Grothendieck's constant [60–62]. This means that both examples show activation of nonlocality of the isotropic state in the range in which it has a projective-LHV model [63]. These examples however do not improve on the $\alpha > \frac{1}{\sqrt{3}}$ visibility achieved via the CHSH inequality. It would be interesting to investigate further if other Bell inequalities (probably with more input settings) could be used to show activation of the isotropic state below this threshold.

## 3.2 Robustness to detection inefficiencies

The isotropic state, $\rho_\alpha = \alpha |\Phi^+\rangle\langle\Phi^+| + (1-\alpha)\frac{\mathbb{1}}{4}$, is a simple model for a noisy quantum state, with $\alpha$ representing the probability of applying a depolarizing channel to one half of a pair of maximally entangled qubits. From an experimental perspective, there are other interesting notions of noise. Notably, detectors are not ideal, and they often fail to register an outcome, opening up loopholes in Bell test experiments. Thus, it is important to study the robustness of nonlocality with respect to detector inefficiencies. Let us first consider the standard Bell scenario and let $\eta$ represent the detection efficiency, the probability of the detector working correctly. We take all detectors to have the same detection efficiency, and assume no detection events of different detectors are statistically independent. Here we consider scenarios with binary inputs taking values in $\{0,1\}$ and binary outcomes taking values in $\{+1,-1\}$. When a no detection event occurs, an outcome in $\{+1,-1\}$ is chosen deterministically as a function of the measurement input of the detector in the round. Mathematically, this is described by two functions $f_A(x), f_B(y) : \{0,1\} \to \pm1$ which give the corresponding outputs given a failure event for each party and their input in that round. Given ideal statistics $p(ab|xy)$ —computed with the noiseless quantum state and measurements— the noisy statistics $P^\eta(a, b|x, y)$ are given by

$$P^\eta(ab|xy) = \eta^2 p(ab|xy) + \eta(1-\eta)\left[\delta_{f_A(x),a}\, p(b|y) + \delta_{f_B(y),b}\, p(a|x)\right]$$
$$+ (1-\eta)^2 \delta_{f_A(x),a}\delta_{f_B(y),b}, \qquad (20)$$

where $\delta_{i,j}$ is the Kronecker delta function. The critical detection threshold $\eta_c$ is defined as the lowest $\eta$ such that, for all $\eta' > \eta$, $P^{\eta'}(a, b|x, y)$ is outside the local set. In this scenario, the best known critical visibility for the two-qubit maximally entangled state is $\eta = 0.8214$ achieved using a Bell inequality with four settings per party [64]. This is very close to the critical detection efficiency of $\eta = 2(\sqrt{2}-1) \simeq 0.8284$ resulting from the CHSH Bell inequality.

Here, we show that by using a single copy of the maximally entangled state in the broadcast scenario, one can achieve a significantly lower critical detection efficiency of $\eta_c = 0.7355$. We consider the tripartite broadcast scenario of Fig. 1a). In this case, we have three measurement devices, and we assume again the same detection efficiency $\eta$ for each device. We similarly consider a strategy in which the detectors output either $\pm 1$ when a failure event occurs, and describe this choice of strategy by three functions $f_A(x), f_B(y), f_C(z) : \{0, 1\} \to \pm 1$. The statistics given a detection efficiency $\eta$ are thus,

$$
\begin{aligned}
P^\eta(abc\,|\,xyz) = {} & \eta^3 p(abc|xyz) \\
& + \eta^2(1-\eta)\big[\delta_{f_A(x),a}\,p(bc|yz) + \delta_{f_B(y),b}\,p(ac|xz) + \delta_{f_C(z),c}\,p(ab|xy)\big] \\
& + \eta(1-\eta)^2\big[\delta_{f_A(x),a}\delta_{f_B(y),b}\,p(c|z) + \delta_{f_A(x),a}\delta_{f_C(z),c}\,p(c|z) + \delta_{f_B(y),b}\delta_{f_C(z),c}\,p(a|x)\big] \\
& + (1-\eta)^3\delta_{f_A(x),a}\delta_{f_B(y),b}\delta_{f_C(z),c}\,.
\end{aligned}
\tag{21}
$$

To find the noiseless statistics that give $\eta_c = 0.7355$, we use the see-saw algorithm from [36, Appx. B]. This algorithm optimizes the robustness of the isotropic state with respect to the visibility parameter $\alpha$ and returns a corresponding Bell inequality valid in the broadcast scenario. We extract the channel and measurements after the algorithm converges and build the ideal statistics $p(abc|xyz)$ using the noiseless isotropic state (i.e. the maximally entangled two-qubit state). Then we find $\eta$ such that $P^\eta(abc|xyz)$ saturates the local bound of the returned Bell inequality, and we do this process over all possible detector strategies $f_A$, $f_B$ and $f_C$. The lowest efficiency found is $\eta_c = 0.7355$, certified by the following inequality

$$
\langle \mathrm{CHSH}[A_0, A_1, B_0, B_1]\,C_1 \rangle - \langle \mathrm{CHSH}[A_0, A_1, B_0, B_1]\rangle + 2(\langle C_1 \rangle - 1) \le 0\,,
\tag{22}
$$

where $\mathrm{CHSH}[A_0, A_1, B_0, B_1] := (A_0 - A_1)B_0 + (A_0 + A_1)B_1$. Here, one adopts a detector failure strategy such that $f_A(x) = -1 \;\forall x, f_B(y) = 1 \;\forall y, f_C(z) = 1 \;\forall z$. We remark that the best detection efficiency is not achieved for the inequality that gives the best visibility (i.e., Eq. (11)). For this inequality, the best critical efficiency we found is $\eta_c = 0.7997$.

We note here that depending on the experimental implementation of the considered scenario, one will not only have inefficiencies coming from detector imperfections (losses, etc.), but also from transmission of the state, and in the case of the broadcast scenario, potential inefficiencies coming from the implementation of the channel are expected.

Previous studies of detector inefficiencies focus on the noise from the detectors and neglect other sources of noise. Therefore, we have adopted the same approach so that we can meaningfully benchmark our results against the existing literature, e.g., Refs. [65–67].

Our result can be compared with the results of [68], where improved bounds are found by using multiple copies of the maximally entangled two-qubit state achieving $\eta_c = 0.8086$, $\eta_c = 0.7399$ and $\eta_c = 0.6929$ for two, three and four copies of the state respectively (and local quantum measurements). Note that that our bound is strictly better than that achieved with three copies of the state, while using a single copy in our scenario (plus a channel). It would be interesting to study if the techniques presented in [68] could be used to find lower efficiency thresholds by applying our strategy in parallel with the use of several copies of the maximally entangled state.

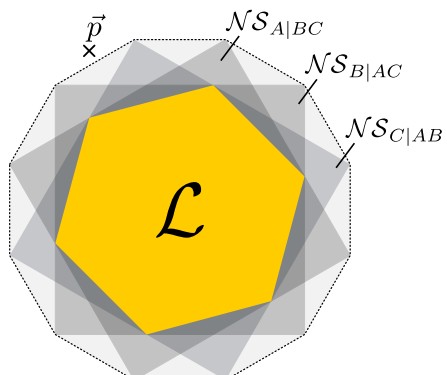

Figure 2: A vector $\vec{p}$ corresponding to a behaviour $\{p(abc|xyz)\}$ is non-signalling bilocal if it is inside the convex hull of all bipartite local polytopes (dashed line), where distant parties respect the non-signalling constraints. Behaviours which are not non-signalling bilocal are referred here as NS genuine tripartite Bell nonlocal. Here, we show that using the broadcast scenario bipartite quantum states with an LHV model for all POVMs can lead to NS genuine tripartite nonlocality.

## 4 Activation of non-signalling genuine multipartite nonlocality

In Refs. [65, 69], the authors analyse different notions of genuine multipartite Bell nonlocality and introduce the concept of non-signalling bilocality, which is intimately related to the idea of broadcast nonlocality presented here. Following Ref. [65], a tripartite behaviour with probabilities $p(abc|xyz)$ is non-signalling bilocal ($NS_2$-local) if it can be written as

$$p(abc|xyz) = q_1 \sum_\lambda \Pi_A(\lambda) p_A(a|x\lambda) p_{BC}^{\mathrm{NS}}(bc|yz\lambda) + q_2 \sum_\lambda \Pi_B(\lambda) p_B(b|y\lambda) p_{AC}^{\mathrm{NS}}(ac|xz\lambda)$$
$$+ q_3 \sum_\lambda \Pi_C(\lambda) p_C(c|z\lambda) p_{AB}^{\mathrm{NS}}(ab|xy\lambda), \tag{23}$$

where all functions $q, p, \pi$ are probability distributions and $p_{AB}^{\mathrm{NS}}(ab|xy\lambda)$, $p_{BC}^{\mathrm{NS}}(bc|yz\lambda)$, $p_{AC}^{\mathrm{NS}}(ac|xz\lambda)$ are non-signalling behaviours. Behaviours that are not $NS_2$-local are then referred to as non-signalling (NS) genuine multipartite nonlocal, see Fig. 2.

We recall from Eq. (8) that a tripartite behaviour with probabilities $p(abc|xyz)$ is broadcast local if it can be written as

$$p(abc|xyz) = \sum_\lambda \Pi_A(\lambda) p_A(a|x\lambda) p_{BC}^{\mathrm{NS}}(bc|yz\lambda). \tag{24}$$

By direct inspection, we then see that any NS genuine multipartite nonlocal behaviour (that is, under definition Eq. (23)) is also broadcast nonlocal. Indeed, the set of non-signalling bilocal behaviours may be viewed as the convex hull of the set of broadcast local in every possible bipartition.

We now present a bipartite state which admits a local hidden-variable model for all possible local measurements but can lead to correlations that are nonlocal in the broadcast scenario. Additionally, we will show that, despite being Bell local in bipartite scenarios, this state displays NS genuine multipartite nonlocality, following the definition of Eq. (23). Consider the following family of two-qubit states:

$$\rho_{\mathrm{POVM}}(\alpha, \chi) := \frac{1}{2}\rho(\alpha, \chi) + \frac{1}{2}\rho_A \otimes |0\rangle\langle 0|, \tag{25}$$

where

$$\rho(\alpha, \chi) := \alpha \left|\psi_\chi\right\rangle\!\left\langle\psi_\chi\right| + (1-\alpha)\frac{\mathbb{1}}{2} \otimes \rho_\chi^B, \tag{26}$$

$$\left|\psi_\chi\right\rangle := \cos\chi\left|00\right\rangle + \sin\chi\left|11\right\rangle, \tag{27}$$

and

$$\rho_\chi^B := \mathrm{Tr}_A\left|\psi_\chi\right\rangle\!\left\langle\psi_\chi\right|, \qquad \rho_A := \mathrm{Tr}_B\,\rho(\alpha, \chi). \tag{28}$$

As shown in Ref. [8], if $\cos^2(2\chi) \geq \frac{2\alpha-1}{(2-\alpha)\alpha^3}$, the state $\rho_{\mathrm{POVM}}(\alpha, \chi)$ admits a local hidden-variable model for all local POVMs performed by Alice and Bob.

We will make use of the inequalities for NS genuine tripartite nonlocality. Reference [65] listed all non-signalling bilocal Bell inequalities in a tripartite scenario where each party has two inputs and two outputs. Using the optimization methods detailed in Appendix B.1 of Ref. [36], we have analysed all these inequalities to check whether there exist a channel $\Omega_{B_0 \to BC}$ and local quantum measurements such that the tripartite state,

$$\rho_{ABC} := \mathbb{1} \otimes \Omega_{B_0 \to BC}[\rho_{\mathrm{POVM}}(\alpha, \chi)], \tag{29}$$

leads to NS-bilocal nonlocality. We have identified that the inequality 16 of Ref. [65],

$$\begin{aligned}
-2\left\langle C_0\right\rangle + \left\langle A_1 B_0\right\rangle + \left\langle A_0 B_1\right\rangle - \left\langle A_0 B_0\right\rangle - \left\langle A_1 B_1\right\rangle + 2\left\langle A_1 C_1\right\rangle + 2\left\langle B_1 C_1\right\rangle \\
+ \left\langle A_1 B_0 C_0\right\rangle - \left\langle A_0 B_0 C_0\right\rangle + \left\langle A_0 B_1 C_0\right\rangle + \left\langle A_1 B_1 C_0\right\rangle \leq 4,
\end{aligned} \tag{30}$$

can be used to show that the state $\rho_{\mathrm{POVM}}(\alpha, \chi)$ is NS-genuine tripartite nonlocal (hence, also broadcast nonlocal) in a region where it admits an LHV model for general POVMs. In Fig. 3 we present the $(\alpha, \chi)$ values for which $\rho_{\mathrm{POVM}}(\alpha, \chi)$ is guaranteed to have a local hidden-variable model (shaded region) and, for each $\chi$, the lowest visibility $\alpha$ for which the state violates an $NS_2$ inequality (dashed blue line). In the intersection of these two regions, The POVM local state (25) can therefore be transformed to a NS genuinely multipartite nonlocal state via the application of a broadcast channel. We note that although standard bipartite nonlocality has been activated from bipartite POVM-local states before, this is the first example of the creation of genuine multipartite nonlocality using a single copy of such a state.

## 4.1 Broadcast activation without a broadcast channel

A reinterpretation of the above results also allows us to construct an example of broadcast activation without a broadcast channel. That is, we consider scenario (a) of Fig. 1, and take the broadcast channel $\Omega_{B_0 \to BC}$ to be the identity channel:

$$\mathbb{1} \otimes \Omega_{B_0 \to BC}[\rho_{AB_0}] = \rho_{AB_0}. \tag{31}$$

Let us now take the state $\rho_{\mathrm{POVM}}$ defined in Eq. (25), in the activation region of Fig. 3 (i.e. anywhere in the yellow-blue intersection). We then apply the channel $\Omega_{B_0 \to BC}$ which leads to activation of NS genuine multipartite nonlocality, the final state thus being

$$\rho_\Omega = \mathbb{1} \otimes \Omega_{B_0 \to BC}[\rho_{\mathrm{POVM}}]. \tag{32}$$

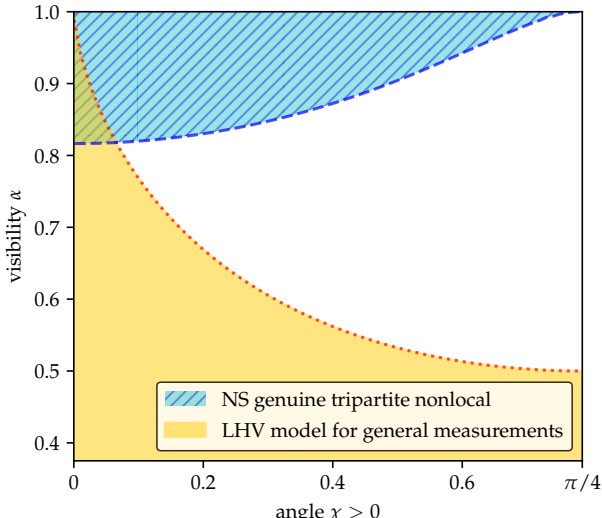

Figure 3: In the shaded (yellow) region, the state $\rho_{\mathrm{POVM}}(\alpha, \chi)$ admits a local hidden-variable model for all POVM measurements. Also, above the (blue) dashed line the state $\rho_{\mathrm{POVM}}(\alpha, \chi)$, with $\chi > 0$, violates the $NS_2$ genuine tripartite inequality number 16 of Ref. [65]. We can see that for small values of $\chi$, there is a range of visibility $\alpha$ such that the state $\rho_{\mathrm{POVM}}(\alpha, \chi)$ is bipartite Bell local in the standard scenario but broadcast nonlocal and NS genuine tripartite nonlocal.

Note that, since local quantum channels cannot create Bell nonlocality from states admitting an LHV model for all POVMs [34],[1] $\rho_\Omega$ has a POVM LHV model on the partition $A|BC$.

$$\rho_\Omega \rightarrow \text{Bell local for all POVMs on } A|BC . \tag{33}$$

However, our previous result shows that $\rho_\Omega$ is NS genuine tripartite nonlocal, and thus broadcast nonlocal too.

$$\rho_\Omega \rightarrow \text{broadcast nonlocal on} A|BC \text{(using scenario (a) of Fig. 1 and inequality (30))} . \tag{34}$$

From this perspective, starting with $\rho_\Omega$ as a bipartite state on $A|BC$ we obtain "activation" by performing the identity channel, i.e. no broadcast channel, and by understanding Bob and Charlie as distinct parties (meaning here that they are restricted to local measurements quantum mechanically and non-signalling strategies classically).

## 5 Device-independent entanglement certification

Here we apply the broadcast scenario to the task of DI entanglement certification. In [36] it was shown that DI entanglement of the two-qubit isotropic state (1) is possible for $\alpha > \frac{1}{2}$, significantly lower than previous best known bound $\alpha \approx 0.6964$ [62]. This result gave promising evidence that DI entanglement certification may be possible in the entire range $\alpha > \frac{1}{3}$ in which the state is entangled. In this section, we give strong evidence this is the case, by showing that DI entanglement certification is possible for $\alpha > 0.338$. To do this, we make use of semi-definite

---

[1]Essentially because applying the dual channel on the measurements (instead of applying the channel on the state) gives rise to the same behaviour, implying a model for all POVMs still holds for that final behaviour.

programming (SDP) tools [70]. As we discuss below, given the proximity of 0.338 to $\frac{1}{3}$, we suspect that this value could be improved to any visibility arbitrarily close to $\frac{1}{3}$ with more computational resources.

The broadcast scenario we consider is the four party scenario shown in Fig. 1b). Each local subsystem of the state of the source is broadcasted to two additional parties. If one considers an arbitrary separable state at the source

$$\rho_{SEP} = \int \Pi(\lambda) \sigma_\lambda^{A_0} \otimes \sigma_\lambda^{C_0} \, d\lambda, \tag{35}$$

with $\Pi(\lambda)$ a normalized probability density, then after the application of the broadcast channels, the most general state shared between the four parties is

$$\rho_{ABCD} = \int \Pi(\lambda) \sigma_\lambda^{AB} \otimes \sigma_\lambda^{CD} \, d\lambda, \tag{36}$$

where $\sigma_\lambda^{AB} = \Omega_{A_0 \to AB}[\sigma_\lambda^{A_0}]$ and $\sigma_\lambda^{CD} = \Omega_{C_0 \to CD}[\sigma_\lambda^{C_0}]$ and $\Omega_{C_0 \to CD}$ and $\Omega_{A_0 \to AB}$ are the quantum channels describing the broadcasting. Local measurements performed on this state lead to behaviours of the form

$$p(abcd|xyzw) = \mathrm{Tr}\left[ \rho_{ABCD} A_{a|x} \otimes B_{b|y} \otimes C_{c|z} \otimes D_{d|w} \right] \tag{37}$$

$$= \int \Pi(\lambda) \mathrm{Tr}\left[ \sigma_\lambda^{AB} A_{a|x} \otimes B_{b|y} \right] \mathrm{Tr}\left[ \sigma_\lambda^{CD} C_{c|z} \otimes D_{d|w} \right] d\lambda \tag{38}$$

$$= \int \Pi(\lambda) \, p_{AB}^Q(ab|xy\lambda) \, p_{CD}^Q(cd|zw\lambda) \, d\lambda, \tag{39}$$

where for each $\lambda$, $p_{AB}^Q(ab|xy\lambda)$ and $p_{CD}^Q(cd|zw\lambda)$ are behaviours from the quantum set of correlations. Alternatively, the behaviours (39) are the most general that can be obtained by making local measurements on a state which is separable with respect to the bipartition AB vs CD. Since these behaviours are the most general that can be obtained from a separable source state, it follows that if a decomposition (39) cannot be found, the source must be entangled, and this therefore constitutes a DI certification of the entanglement of the source.

The question remains, however, of how to prove that a given behaviour does not admit a decomposition (39). This is complicated by the fact that the states and the measurements can in principle act on infinite dimensional Hilbert spaces. In order to tackle this, we will make use of a semi-definite programming technique introduced in [71] and based on the NPA hierarchy [72]. For a fixed number of inputs and outputs, let us denote the set of behaviours admitting a decomposition (39) by $\mathcal{Q}_{AB|CD}$ so that $p \in \mathcal{Q}_{AB|CD}$ if and only if (39) is satisfied. Furthermore, let us denote by $\mathcal{Q}_{AB|CD}^{PPT}$ the set of correlations[2] obtained by using states that admit a positive partial transpose (PPT) with respect to the bipartition AB vs CD. Since separable states are PPT, it follows that $\mathcal{Q}_{AB|CD} \subseteq \mathcal{Q}_{AB|CD}^{PPT}$ and thus $p \notin \mathcal{Q}_{AB|CD}^{PPT} \implies p \notin \mathcal{Q}_{AB|CD}$. A certificate that $p \notin \mathcal{Q}_{AB|CD}^{PPT}$ is therefore a device-independent proof of entanglement.

We now describe how one can obtain such a certificate. In [71] it is shown that the set $\mathcal{Q}_{AB|CD}^{PPT}$ can be characterized from the outside by a sequence of semidefinite programs. The first and simplest SDP in this sequence is constructed as follows. One defines a set

$$
\begin{aligned}
\mathcal{S}_1 = \{ & \{\tilde{A}_{a|x}\}_{a,x}, \{\tilde{B}_{b|y}\}_{b,y}, \{\tilde{C}_{c|z}\}_{c,z}, \{\tilde{D}_{d|w}\}_{d,w}, \\
& \{\tilde{A}_{a|x}\tilde{B}_{b|y}\}_{a,b,x,y}, \{\tilde{A}_{a|x}\tilde{C}_{c|z}\}_{a,c,x,z}, \ldots, \{\tilde{C}_{c|z}\tilde{D}_{d|w}\}_{c,d,z,w}, \{\tilde{A}_{a|x}\tilde{B}_{b|y}\tilde{C}_{c|z}\}_{a,b,c,x,y,z}, \ldots, \\
& \{\tilde{B}_{b|y}\tilde{C}_{c|z}\tilde{D}_{d|w}\}_{b,c,d,y,z,w}, \{\tilde{A}_{a|x}\tilde{B}_{b|y}\tilde{C}_{c|z}\tilde{D}_{d|w}\}_{a,b,c,d,x,y,z,w} \}, \tag{40}
\end{aligned}
$$

---

[2]To be more precise, for both $\mathcal{Q}_{AB|CD}$ and $\mathcal{Q}_{AB|CD}^{PPT}$ we assume that a behaviour is generated by a 'commuting operator' strategy, *i.e.* where measurement operators for different parties are assumed to commute, but a tensor product structure does not necessarily hold.

consisting of arbitrary measurement operators for the four parties. Measurement operators for different parties commute since they act on different Hilbert spaces, *e.g.*,

$$[\tilde{A}_{a|x}, \tilde{B}_{b|y}] = 0 \,. \tag{41}$$

Since the Hilbert space dimension is unbounded, one can assume the measurements are projective without loss of generality, *e.g.*,

$$\tilde{A}_{a|x}\tilde{A}_{a'|x} = \tilde{A}_{a|x}\delta_{a,a'} \,, \tag{42}$$

and similarly for the other parties. A matrix $\Gamma$, called the moment matrix, is then constructed, with elements

$$\Gamma^{i,j} = \mathrm{Tr}(\rho_{ABCD}E_i E_j^\dagger), \tag{43}$$

and $E_i, E_j \in \mathcal{S}_1$. The matrix $\Gamma$ can be shown to be positive semi-definite, and satisfies a number of linear constraints that follow from (41) and (42). Furthermore, from the Born rule, some elements correspond to observable probabilities $p(abcd|xyzw)$. Finally, as operators acting on $\mathcal{S}_1$ preserve the structure of each local system, operations with respect to each subsystem can be applied to the moment matrix. For example, performing a partial transpose of $\rho_{ABCD}$ results in a corresponding partial transposition defined on the moment matrix $\Gamma$ [71]. Importantly, this implies that if $\rho_{ABCD}$ is PPT with respect to $AB|CD$, a corresponding PPT constraint also holds for $\Gamma$. Any $p \in \mathcal{Q}_{AB|CD}^{PPT}$ therefore implies the existence of a matrix $\Gamma$ with the above constraints.

For a given set of probabilities $p(abcd|xyzw)$, these constraints give a necessary condition for $p \in \mathcal{Q}_{AB|CD}^{PPT}$. Namely, if $p \in \mathcal{Q}_{AB|CD}^{PPT}$ then there must exist a way of completing the matrix $\Gamma$ (having fixed the elements corresponding to the probabilities), such that $\Gamma \succeq 0$, $\Gamma$ satisfies the linear constraints implied by (41) and (42), and $\Gamma$ is PPT in the sense described in [71]. Conversely, if such a completion cannot be found, then the state $\rho_{AB|CD}$ cannot be PPT, and is therefore entangled. Note that since these constraints are linear and semi-definite with respect to $\Gamma$, this can be cast as an instance of a semi-definite program, which in the case of infeasibility, returns a numerical certificate that $p \notin \mathcal{Q}_{AB|CD}$, *i.e.*, that certifies the entanglement of the state.

Using the above, we were able to prove device-independent entanglement certification for $\rho_\alpha$ for $\alpha > 0.338$. To achieve this, we used a heuristic optimization procedure described in Appendix A. The precise strategy involves each of the parties making one of three measurements. The numerical values of the measurement and channels, as well as the corresponding Bell inequality that certifies this visibility, can be found in the GitHub repository for the article. Although we could not obtain a proof that $\alpha > \frac{1}{3}$ implies the possibility of a DI entanglement certification in the broadcast scenario, our numerical analysis strongly suggests that all entangled two-qubit Werner states can be DI certified in the broadcast scenario. This result suggests that an analytic proof of DI entanglement certification for $\alpha > \frac{1}{3}$ may be within reach, and would be an exciting avenue of future research. Given that this is exactly the separability limit, one may even hope that broadcasting could activate DI entanglement cert for all entangled states.

## 6 Broadcast steering

In this section, we introduce a new broadcast scenario which is based on EPR-steering from Bob to Alice. This represents a scenario where the measurements performed by Alice are completely characterised, but no assumption is made on the measurements made by Bob. In this broadcast steering scenario we present a novel example of activation of EPR-steering correlations, which does not rely on previous methods such as local filtering [35] or the multi-copy regime [41].

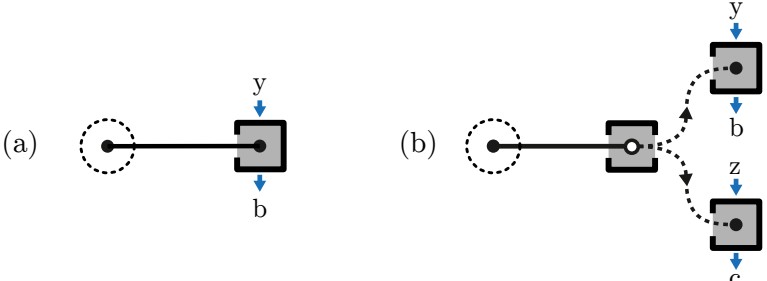

Figure 4: Illustration of a steering scenario where the measurements performed by Alice are trusted to be completely characterized (hence, represented by a circle). Figure a) represents the standard steering scenario and b) represents the case where a broadcast channel sends the system to two parties which only have access to non-signalling resources.

## 6.1 Standard quantum steering

Before presenting the EPR-steering broadcast scenario, we review the concept of standard EPR-steering [3]. For a more detailed introduction, we recommend the review articles [4, 5]. We consider a scenario where Alice and Bob share a bipartite state $\rho$ and Bob has access to a set of local POVMs described by $\{B_{b|y}\}$. When Bob performs the measurement labelled by $y$ and obtains the outcome $b$, the physical system held by Alice is described by its *assemblage*, a set of unnormalized states defined by

$$\sigma_{b|y} := \mathrm{Tr}_B(\mathbb{1} \otimes B_{b|y}\rho), \tag{44}$$

where $\mathrm{Tr}_B$ denotes the partial trace over Bob's subsystem and the $\mathrm{Tr}(\sigma_{b|y})$ corresponds to the probability of obtaining the output $b$ given input $y$ for Bob. An assemblage admits a local hidden-state (LHS) model if it can be written as

$$\sigma_{b|y} = \int \Pi(\lambda)\, \sigma_\lambda\, p_B(b|y, \lambda)\, d\lambda, \tag{45}$$

where $\lambda$ stands for a hidden variable and $\{\Pi(\lambda)\}_\lambda$ and $\{p_B(b|y, \lambda)\}_b$ are probability distributions. We thus say that an assemblage $\sigma_{b|y}$ is *steerable* if it does not admit an LHS decomposition of the form (45).

## 6.2 Steering in the broadcast scenario

In the simplest broadcast scenario, we start with bipartite state $\rho_{AB_0}$, which after channel $\Omega_{B_0 \to BC}$ is mapped to state $\rho_{ABC}$, shared by Alice, Bob and Charlie (who can perform local measurements $x, y, z$, with respective outcomes $a, b, c$). The question is then whether the statistics observed by Alice can be explained by a local hidden-state model. That is, can the assemblage

$$\sigma_{bc|yz} = \mathrm{Tr}_{BC}\left(\mathbb{1} \otimes B_{b|y} \otimes C_{c|z}\left[\mathbb{1} \otimes \Omega_{B_0 \to BC}(\rho)\right]\right), \tag{46}$$

be written as

$$\sigma_{bc|yz} = \int \Pi(\lambda)\, \sigma_\lambda\, p_{BC}^{\mathrm{NS}}(b, c|y, z, \lambda)\, d\lambda, \tag{47}$$

where $p_{BC}^{\mathrm{NS}}(b, c|y, z, \lambda)$ is an arbitrary non-signalling behaviour between Bob and Charlie, for each value $\lambda$. We refer to a violation of (47) as *broadcast steering*, and similarly to broadcast

nonlocality, such a violation cannot be explained by the transformation device alone, as long as it generates non-signalling resources only.

In this work, we also consider the scenario where the broadcast channel $\Omega$ maps the space $B_0$ to a tripartite space $B \otimes C \otimes D$. That is, after the channel the state is a four-partite state $\rho_{ABCD}$, shared between Alice, Bob, Charlie and Dave. We then consider an assemblage

$$\sigma_{bcd|yzw} = \text{Tr}_{BCD}\left(\mathbb{1} \otimes B_{b|y} \otimes C_{c|z} \otimes D_{d|w}\left[\mathbb{1} \otimes \Omega_{B_0 \to BCD}(\rho)\right]\right), \qquad (48)$$

which is broadcast steerable if it can be written as

$$\sigma_{bcd|yzw} = \int \Pi(\lambda)\, \sigma_\lambda\, p_{BCD}^{\text{NS}}(b, c, d|y, z, w\lambda)\, d\lambda, \qquad (49)$$

where $p_{BCD}^{\text{NS}}(b, c, d|y, z, w\lambda)$ is an arbitrary non-signalling behaviour.

Before finishing this subsection, we remark that, in a standard steering scenario (steering from Bob to Alice), the main hypothesis is that Alice's measurements are trusted to be fully characterized. In the broadcast steering case, we need an additional hypothesis, which is that Bob and Charlie are restricted to non-signalling resources, that is, they may be strongly correlated, but they cannot communicate. Nevertheless, this hypothesis can be imposed in a physical and fair way by ensuring that the parties after the broadcast channel are in space-like separated areas at the time of the measurements.

### 6.2.1 Other potential notions of broadcast steering

We presented broadcast steering for the case where Alice is the trusted party. In principle, one could consider other natural configurations for defining broadcast steering:

- Bob and Charlie are the trusted parties: in that case, one wonders whether the assemblage

$$\sigma_{a|x} = \text{Tr}_A(A_{a|x} \otimes \mathbb{1}_B \otimes \mathbb{1}_C[\mathbb{1} \otimes \Omega_{B_0 \to BC}(\rho)]), \qquad (50)$$

  can be written as

$$\sigma_{a|x} = \int \Pi(\lambda)\, \sigma_\lambda\, p_A(a|x, \lambda)\, d\lambda. \qquad (51)$$

  Note however that, this corresponds to standard steering, where Bob and Charlie can be seen as a single party. Thus, this scenario is trivial and no activation is possible in this case.

- Hybrid cases: either Alice and Bob are trusted, or only Charlie is trusted. In either case, the absence of an LHS model does not imply anything about $\rho_{AB_0}$, since it could be explained by (standard) bipartite steering between Bob and Charlie.

Since these two other approaches lead to trivial definitions, we focus on the definition described by (47) where only Alice performed trusted and characterized measurements.

### 6.3 Broadcast steering with the two-qubit isotropic state

We now present some steering activation results in broadcast scenarios by carefully analysing the two-qubit isotropic state:[3]

$$\rho_\alpha = \alpha|\phi^+\rangle\langle\phi^+| + (1-\alpha)\frac{\mathbb{1}}{4}. \qquad (52)$$

---

[3]We remark that since the two-qubit isotropic state is local-unitary equivalent to the two-qubit Werner state, all results presented in this subsection also hold for the two-qubit Werner state.

The isotropic state represents a maximally entangled state which undergoes white noise. Due to its symmetry, simplicity, and experimental relevance, the isotropic state is often used as a benchmark for several tasks in quantum information. Up to the moment of writing this manuscript, in the standard steering scenario, the two-qubit isotropic state was only shown to be steerable for visibility $\alpha > \frac{1}{2}$ [3]. Moreover, the two-qubit isotropic state has an LHS model for projective measurements when $\alpha \leq \frac{1}{2}$ [3], and there is evidence that it also has an LHS model for general POVMs when $\alpha \leq \frac{1}{2}$ [73, 74]. The results presented in this subsection were obtained with the help of the heuristic search described in the Appendix C found in the GitHub online repository [75].

*Two broadcasted parties*—We first consider a scenario where there are two parties, Bob and Charlie after the broadcast channel as in Fig. 1a). When Bob and Charlie can choose between two dichotomic measurements each, that is, $y \in \{0, 1\}$ and $z \in \{0, 1\}$, we could find a channel $\Omega$ and measurements $\{B_{b|y}\}$, $\{C_{c|z}\}$, to certify broadcast steering for $\alpha > 0.5616$. We also investigated the scenario where Bob and Charlie have access to three dichotomic measurements each, *i.e.*, $y \in \{0, 1, 2\}$ and $z \in \{0, 1, 2\}$. In this case, we detected broadcast steering up to $\alpha > 0.4945$.

*Three broadcasted parties*—We now consider the scenario where there are three parties, Bob, Charlie and Dave, after the broadcast channel as in Fig. 1c). We focused on the scenario where Bob, Charlie and Dave can choose between two dichotomic measurements. Since the vertices of the non-signalling polytope for three parties performing two dichotomic measurements were explicitly obtained at Ref. [76], we can use these vertices to run (a straightforward extension of) our heuristic procedure presented in Appendix C. This allowed us to certify that the two-qubit isotropic state exhibits broadcast steering for $\alpha > 0.4678$, showcasing an even stronger example of steering activation.

Note that the latter two results are example of activation of steering (relative to projective measurements), since the isotropicc state admits a LHS model in the range $\alpha \leq \frac{1}{2}$. These are the first examples of single-copy activation of steering for this class of states.

# 7 Discussion

The relationship between quantum entanglement and Bell nonlocality plays a major role in understanding quantum correlations and the development of device-independent protocols. In a seminal paper, Werner showed that entangled states may admit a local hidden-variable model and cannot lead to Bell nonlocal correlations in the standard Bell scenario [33]. What seemed to be definite proof that some entangled states cannot lead to nonlocality is today recognized as only a first (fundamental) step. Over the past years, natural extensions of Bell scenarios revealed that states admitting local hidden-variable models may also display nonlocal correlations [37] and we are forced to accept that the relationship between entanglement and nonlocality is far from being fully understood.

This work investigates entanglement and nonlocality scenarios where the parties can broadcast their systems to reveal strong correlations which are hidden in the standard Bell test. From a foundational perspective, we provided novel examples of how to activate the nonlocality of entangled states which admit local hidden-variable models. We presented an example of bipartite local states leading to genuine multipartite nonlocality, introduced the concept of broadcast device-independent entanglement certification and the concept of broadcast steering. From a more practical aspect, we developed analytical and computational methods to analyse entanglement and nonlocality in broadcast scenarios. Our findings advance the discussion on whether entanglement can lead to nonlocality, and we hope that the methods presented here may pave the way for network-based and broadcast-based device-independent protocols.

All our code can be found in the GitHub online repository [75] and can be freely used under the MIT licence.[4]

## Acknowledgments

We thank Máté Farkas for interesting discussions on detection efficiencies.

M.T.Q. acknowledges the Austrian Science Fund (FWF) through the SFB project BeyondC (sub-project F7103), a grant from the Foundational Questions Institute (FQXi) as part of the Quantum Information Structure of Spacetime (QISS) Project (qiss.fr). The opinions expressed in this publication are those of the authors and do not necessarily reflect the views of the John Templeton Foundation. This project has received funding from the European Union's Horizon 2020 research and innovation programme under the Marie Skłodowska-Curie grant agreement No 801110. It reflects only the authors' view, the EU Agency is not responsible for any use that may be made of the information it contains. ESQ has received funding from the Austrian Federal Ministry of Education, Science and Research (BMBWF).

E-CB. and JB acknowledge support from Fundació Cellex, Fundació Mir-Puig, and from Generalitat de Catalunya through the CERCA program. JB acknowledges support from the AXA Chair in Quantum Information Science.E-CB. acknowledges financial support from the Spanish State Research Agency through the "Severo Ochoa" program for Centers of Excellence in R&D (CEX2019-000910-S), from Fundació Cellex, Fundació Mir-Puig, and from Generalitat de Catalunya through the CERCA program. This project has received funding from the "Presidencia de la Agencia Estatal de Investigación" within the "Convocatoria de tramitación anticipada, correspondiente al año 20XX, de las ayudas para contratos predoctorales (Ref. PRE2019-088482) para la formación de doctores contemplada en el Subprograma Estatal de Formación del Programa Estatal de Promoción del Talento y su Empleabilidad en I+D+i, en el marco del Plan Estatal de Investigación Científica y Técnica y de Innovación 2017-2020, cofinanciado por el Fondo Social Europeo."

F.H. acknowledges funding from the Swiss National Fund (SNF) through the Postdoc Mobility fellowship P400P2_199309.

## A Heuristic method for device-independent entanglement certification

To search for a witness for $\rho_{\alpha > 0.338}$, we employ the following heuristic optimization.

1. Pick random (projective) measurements $\{A_{a|x}\}$, $\{B_{b|y}\}$, $\{C_{c|z}\}$, $\{D_{d|w}\}$, and channels $\sigma_\lambda^{AB} = \Omega_{A_0 \to AB}[\sigma_\lambda^{B_0}]$, $\sigma_\lambda^{CD} = \Omega_{C_0 \to CD}[\sigma_\lambda^{C_0}]$.

2. Find $\alpha^*$ such that the resulting correlation from $\rho_\alpha$ is on the boundary of $\mathcal{Q}_{AB|CD}^{PPT,1}$. This can be done via a semi-definite programming described below.

3. Extract corresponding inequality $F$.

4. For state $\rho_{\alpha^*}$, optimize the inequality $F$ over all POVMs $\{A_{a|x}\}$, $\{B_{b|y}\}$, $\{C_{c|z}\}$, $\{D_{d|w}\}$ and channels $\sigma_\lambda^{AB} = \Omega_{A_0 \to AB}[\sigma_\lambda^{B_0}]$, $\sigma_\lambda^{CD} = \Omega_{C_0 \to CD}[\sigma_\lambda^{C_0}]$.

5. Repeat point 2-4 until two successive values of $\alpha^*$ are identical.

---

[4]https://opensource.org/licenses/mit

In order to find the value such that $\rho_{\alpha^*}$ is on the boundary of $\mathcal{Q}_{AB|CD}^{PPT,1}$, one can run the following SDP

$$\text{maximise } \alpha,$$
$$\text{s.t. } \Gamma(p(abcd|xyzw)) \succeq 0,$$

where $p(abcd|xyzw) = \text{Tr}\left[\Omega_{A_0 \to AB} \cdot \Omega_{C_0 \to CD}(\rho_\alpha) A_{a|x} \otimes B_{b|y} \otimes C_{c|z} \otimes D_{d|w}\right]$. Here, the moment matrix $\Gamma$ represents the characterization of $\mathcal{Q}_{AB|CD}^{PPT,1}$. In principle, one may consider a tighter approximation than the first level of the Moroder *et. al.* hierarchy by employing the corresponding moment matrix in the above optimization.

# B  Efficient method for computing the LHS bound of steering inequalities

We now describe the method for computing LHS bounds of steering inequalities used in our work. A similar formula was previously used in Refs. [78] and [79]. We derive a proof here for completeness:

For an assemblage, $\sigma_{a|x}$ a steering inequality is of the form:

$$\sum_{a,x} \text{Tr}(F_{a|x}\sigma_{a|x}) \leq L, \tag{53}$$

where $F_{a|x}$ are matrices of the same dimension of $\sigma_{a|x}$, and $L$ is the LHS bound of the inequality, that is, the maximal value attained with an LHS assemblage. Formally:

$$L = \max_{\sigma_\lambda}\left\{\sum_{a,x} \text{Tr}(F_{a|x}\sigma_{a|x}) \,\middle|\, \sigma_{a|x} = \sum_\lambda \sigma_\lambda D^{(\lambda)}(a|x),\ \sigma_\lambda \geq 0,\ \text{Tr}\left(\sum_\lambda \sigma_\lambda\right) = 1\right\}, \tag{54}$$

where $\lambda$ runs over all deterministic strategies $D^{(\lambda)}(a|x)$. From equation (54), one can see that the LHS bound $L$ can be computed with an SDP optimization (linear objective function and SDP conditions of the variables $\sigma_\lambda$). However, one can devise a more efficient formula to compute it. Let us define $M_k := \sum_{a,x} F_{a|x} D^{(k)}(a|x)$ and consider the inequality applied to an unsteerable assemblage:

$$\sum_{a,x} \text{Tr}\left(F_{a|x}\sigma_{a|x}^{\text{LHS}}\right) = \sum_{a,x} \text{Tr}\left(F_{a|x}\sum_k \sigma_k D^{(k)}(a|x)\right) \tag{55}$$

$$= \sum_k \text{Tr}\left(\sigma_k \sum_{a,x} F_{a|x} D^{(k)}(a|x)\right) \tag{56}$$

$$= \sum_k \text{Tr}(\sigma_k M_k) = \sum_k p_k \text{Tr}(\hat{\sigma}_k M_k) \tag{57}$$

$$\leq \max_k \text{Tr}(\hat{\sigma}_k M_k) \leq \max_k \lambda_M(M_k), \tag{58}$$

where $\hat{\sigma}$ means $\text{Tr}(\hat{\sigma}) = 1$ and $\lambda_M(A)$ means the largest eigenvalue of $A$. Moreover, one can see that the bound is tight (it can be achieved by setting all $\sigma_k$ to 0 but the one corresponding to the $M_k$ with the maximal largest eigenvalue, which is set to the projector onto the corresponding eigenvector). All in all we have that

$$L = \max_k \lambda_M(M_k). \tag{59}$$

## C  Heuristic search for certifying broadcast steering of bipartite states

Here we describe how we searched for interesting examples of steering in the broadcast scenario. For convenience, we consider the scenario featuring 3 parties, see scenario b) of figure 4. Note however that it extends straightforwardly to more parties. Let us consider a family of state of the form

$$\rho_v = v\rho_{NL} + (1-v)\rho_{SEP}, \tag{60}$$

where $\rho_{NL}$ is typically a Bell nonlocal state while $\rho_{SEP}$ is separable, and the linear parameter $0 \leq v \leq 1$. For example, the isotropic state of two qubits is of that form:

$$\rho_\alpha = \alpha|\phi^+\rangle\langle\phi^+| + (1-\alpha)\frac{\mathbb{1}}{4}. \tag{61}$$

Here the goal is to find the smallest possible $v$ such that the state exhibits broadcast steering. We used the following procedure

1. Pick random (projective) measurements $\{B_{b|y}\}$, $\{C_{c|z}\}$ and channel $\Omega_{B_0 \to BC}$.

2. Find $v^*$ such that the resulting assemblage using state $\rho_v$ is broadcast steerable. This can be done via a semi-definite programming described below.

3. Extract corresponding steering inequality $F$.

4. For state $\rho_{v^*}$, optimize steering inequality $F$ over all POVMs $\{B_{b|y}\}$, $\{C_{c|z}\}$ and channels $\Omega_{B_0 \to BC}$.

5. Repeat point 2-4 until two successive values of $v^*$ are identical.

In order to find the value such that $\rho_v$ is broadcast steerable for fixed measurements and channel (step 2), one can run the following SDP

$$\text{maximise } v,$$
$$\text{s.t. } \text{Tr}_{BC}(\mathbb{1} \otimes B_{b|y} \otimes C_{c|z}[\mathbb{1} \otimes \Omega_{B_0 \to BC}(\rho_v)]) = \sum_k \sigma_k D_{NS}^{(k)}(b, c|y, z),$$
$$\sigma_k \geq 0, \text{Tr}(\sum_k \sigma_k) = 1,$$

where the $\sigma_k$ (together with $v$) are the SDP variables and the $D_{NS}^{(k)}(b, c|y, z)$ are the extremal non-signalling strategies between Bob and Charlie. The dual variables of the equality constraints of this SDP provide a witness $F$, that is, a steering inequality of the form

$$\sum_{a,x} \text{Tr}(F_{a|x}\sigma_{a|x}) \leq L, \tag{62}$$

here $F_{a|x}$ are matrices of the same dimension of $\sigma_{a|x}$, and $L$ is the LHS bound of the inequality, that is, the maximal value attained with an LHS assemblage. Formally:

$$L = \max_{\sigma_\lambda} \left\{ \sum_{a,x} \text{Tr}(F_{a|x}\sigma_{a|x}) \,\middle|\, \sigma_{a|x} = \sum_\lambda \sigma_\lambda D^{(\lambda)}(a|x), \ \sigma_\lambda \geq 0, \ \text{Tr}\left(\sum_\lambda \sigma_\lambda\right) = 1 \right\}, \tag{63}$$

where $\lambda$ runs over all deterministic strategies $D^{(\lambda)}(a|x)$. A formula to compute the LHS bound $L$ of such an inequality is given in Appendix B. An algorithm to maximize such a steering inequality (step 4) over measurements and channels is given in Appendix C.1.

### C.1 Optimizing a steering inequality

Assume one wants to maximize the violation of a steering inequality characterized by operators $F_{bc|yz}$ for a fixed state $\rho_{AB_0}$ and over channels $\Omega_{B_0 \to BC}$ and POVMs $\{B_{b|y}\}$, $\{C_{c|z}\}$. This means one wants to maximize:

$$\mathrm{Tr}\left[\sum_{b,c,y,z} F_{bc|yz} \mathrm{Tr}_{B_0 BC}(\mathbb{1}_A \otimes B_{b|y} \otimes C_{c|z} [\mathbb{1}_A \otimes \Omega_{B_0 \to BC}(\rho_{AB_0})])\right]. \tag{64}$$

Both the objective function and the constraints on the variables are thus nonlinear, making the naive parametrization and optimization potentially inefficient. One can instead decompose the optimization on several subsets of variables, such that each optimization can be performed efficiently (aka see-saw optimization). Here, we used the following procedure:

1. Fix randomly POVMs $\{C_{c|z}\}$ and channel $\Omega_{B_0 \to BC}$.

2. Optimize the inequality with respect to POVMs $\{B_{b|y}\}$, update variables accordingly.

3. Optimize the inequality with respect to POVMs $\{C_{c|z}\}$, update variables accordingly.

4. Optimize the inequality with respect to channels $\Omega_{B_0 \to BC}$, update variables accordingly.

5. Repeat point 2 - 4 until two successive values of the inequality are equal (up to some desired precision).

The motivation for such a heuristic is that steps 2-4 can be written as single-shot SDPs. Indeed, for step 2 the constraints are $B_{b|y} \geq 0$ and $\sum_b B_{b|y} = \mathbb{1}$, and the objective function is linear. Step 3 is similar. For step 4, we can use the Choi-Jamiolkowski isomorphism [77]: the action of the map $\Omega_{B_0 \to BC}$ on some state $\sigma_{B_0}$ can be written as

$$\Omega_{B_0 \to BC}(\sigma_{B_0}) = \mathrm{Tr}_1(\rho_\Omega(\sigma_{B_0}^T \otimes \mathbb{1}_{BC})), \tag{65}$$

where $\rho_\Omega \equiv d \cdot \mathbb{1} \otimes \Omega[|\Phi^+\rangle\langle\Phi^+|]$ is called the Choi state of the map $\Omega_{B_0 \to BC}$ (where $|\Phi^+\rangle$ is the maximally entangled state of local dimension $d = \dim(\mathcal{H}_{B_0})$).

For valid channels, the Choi state satisfies $\rho_\Omega \geq 0$ and $\mathrm{Tr}_{BC}(\rho_\Omega) = \mathbb{1}_{B_0}$. The Choi-Jamiolkowski isomorphism ensures that for each state satisfying these two constraints, there is a unique corresponding channel. We can thus use the variable $\rho_\Omega$, which can be treated as an SDP variable, to solve step 4. One can indeed write the steering inequality as a linear function of $\rho_\Omega$:

$$\mathrm{Tr}\left[\sum_{b,c,y,z} F_{bc|yz} \mathrm{Tr}_{B_0 BC}((\mathbb{1}_A \otimes \mathbb{1}_{B_0} \otimes B_{b|y} \otimes C_{c|z})(\mathbb{1}_A \otimes \rho_\Omega)(\rho_{AB_0}^{T_{B_0}} \otimes \mathbb{1}_{BC}))\right]. \tag{66}$$

Therefore, each step of the aforementioned procedure can be efficiently carried, since single-shot SDPs provide global optimums in polynomial time. In practice, we indeed observe that the entire see-saw optimization converges to what seems to be the global maximum in a few dozens seconds, for two-qubit states on bipartite and tripartite broadcast steering scenarios.

# D   Proof of the lifting *ansatz* in Ineq. (17)

We rewrite Ineq. (17) for convenience:

$$\langle \mathcal{I}[A_0, \ldots, A_m, C_0, \ldots, C_k](B_0 + B_1)\rangle + \mathcal{L}_\mathcal{I}\langle A_{m+1}(B_1 - B_0)\rangle \leq 2\mathcal{L}_\mathcal{I}. \tag{67}$$

To prove it, we follow the same logic as in [36, Sec. 4.1]. For the set of broadcast local distributions, the extremal strategies[5] consist of a deterministic strategy for Alice (this already implies $\langle A_x B_y C_z\rangle = \langle A_x\rangle\langle B_y C_z\rangle$), and, for Bob and Charlie either a local deterministic strategy or a nonlocal extremal strategy.

Assuming a local deterministic strategy for Bob and Charlie, this further implies $\langle B_y C_z\rangle = \langle B_y\rangle\langle C_z\rangle$. Ineq. (17) becomes:

$$\langle \mathcal{I}[A_0, \ldots, A_m, C_0, \ldots, C_k]\rangle\langle B_0 + B_1\rangle + \mathcal{L}_\mathcal{I}\langle A_{m+1}\rangle\langle B_1 - B_0\rangle \leq 2\mathcal{L}_\mathcal{I}.$$

For any deterministic strategy, the values of the 1-body correlators are extremal, *i.e.*, $\langle B_y\rangle, \langle C_z\rangle \in \{+1, -1\}$. As such, either $\langle B_0 + B_1\rangle = 0$ and $\langle B_0 - B_1\rangle = \pm 2$, or vice versa. Assuming the first case, then the first term is zero, and it is direct to see that the bound is satisfied. It is also easy to check that this is true for the other case.

Assume now a nonlocal extremal strategy for Bob and Charlie. The correlators in the second term do not involve Charlie, and as such, it factorizes as follows:

$$\mathcal{L}_\mathcal{I}\langle A_{m+1}(B_1 - B_0)\rangle = \mathcal{L}_\mathcal{I}\langle A_{m+1}\rangle(\langle B_1\rangle - \langle B_0\rangle).$$

It has been shown [80, Table. II] that for any 2-output non-local extremal distribution with 2 inputs for one party and any number of inputs for the other party, the marginals for the party with 2 inputs are all equal to $\frac{1}{2}$. This means that $\langle B_0\rangle = \langle B_1\rangle = 0$, which implies that the second term is zero.

Regarding the first term, expand $\mathcal{I}$ as a linear combination of correlators:

$$\langle \mathcal{I}[A_0, \ldots, A_m, C_0, \ldots, C_k](B_0 + B_1)\rangle$$
$$= \sum_{i=0}^{m}\sum_{j=0}^{k} M_{ij}\langle A_i\rangle\langle(B_0 + B_1)C_j\rangle + \sum_{i=0}^{m}\nu_i\langle A_i\rangle(\langle B_0\rangle + \langle B_1\rangle) + \sum_{j=0}^{k}\mu_j\langle(B_0 + B_1)C_j\rangle. \tag{68}$$

Notice that since $\langle B_0\rangle + \langle B_1\rangle = 0$, any contribution from the $\langle A_i\rangle$ terms in the inequality vanishes, which might affect the bound of the inequality. From henceforth, assume that $\mathcal{I}$ contains no 1-body correlator terms for Alice (i.e., $\nu_i = 0$). Then we can absorb $B_0$ and $B_1$ into the $C$'s in the following sense:

$$\langle \mathcal{I}[A_0, \ldots, A_m, C_0, \ldots, C_k](B_0 + B_1)\rangle = \langle \mathcal{I}[A_0, \ldots, A_m, B_0 C_0, \ldots, B_0 C_k]\rangle$$
$$+ \langle \mathcal{I}[A_0, \ldots, A_m, B_1 C_0, \ldots, B_1 C_k]\rangle.$$

Now, from $\langle A_x B_y C_z\rangle = \langle A_x\rangle\langle B_y C_z\rangle$ and since $-1 \leq \langle A_x\rangle \leq 1$ and $-1 \leq \langle B_y C_z\rangle \leq 1$ one has

$$\langle \mathcal{I}[A_0, \ldots, A_m, B_y C_0, \ldots, B_y C_k]\rangle \leq \max_{|\langle A_x\rangle|, |\langle B_y C_z\rangle|\leq 1} \mathcal{I}[\langle A_0\rangle, \ldots, \langle A_m\rangle, \langle B_y C_0\rangle, \ldots, \langle B_y C_k\rangle]$$
$$= \max_{|\langle A_x\rangle|, |\langle C_z\rangle|\leq 1} \mathcal{I}[\langle A_0\rangle, \ldots, \langle A_m\rangle, \langle C_0\rangle, \ldots, \langle C_k\rangle]$$
$$= \mathcal{L}_\mathcal{I}, \tag{69}$$

which implies the bound of $2\mathcal{L}_\mathcal{I}$.

---

[5]We use "probability distribution", "strategy" and "behaviour" interchangeably.

This concludes the proof of equation (17). The other lifting *ansatz* concerns the 4-partite symmetric broadcast scenario:

$$\langle \mathcal{I}[A_0, \dots, A_m, C_0, \dots, C_k](B_0 + B_1) D_0 \rangle + \mathcal{L}_{\mathcal{I}} \langle (B_1 - B_0) D_1 \rangle \leq 2\mathcal{L}_{\mathcal{I}}.$$

If everyone has a local deterministic strategy, then the expression simplifies to

$$\langle \mathcal{I}[A_0, \dots, A_m, C_0, \dots, C_k] \rangle \langle B_0 + B_1 \rangle \langle D_0 \rangle + \mathcal{L}_{\mathcal{I}} \langle B_1 - B_0 \rangle \langle D_1 \rangle.$$

Since either $\langle B_0 + B_1 \rangle = \pm 2$ and $\langle B_1 - B_0 \rangle = 0$ or vice-versa, the $2\mathcal{L}_{\mathcal{I}}$ bound follows using the reasoning from the previous proof in this section.

If Alice and Bob share a NS resource and Charlie and Dave do a local deterministic strategy, then the second term in the inequality is zero because $\langle B_0 \rangle = \langle B_1 \rangle = 0$. The first term simplifies to

$$\langle \mathcal{I}[A_0(B_0 + B_1), \dots, A_m(B_0 + B_1), C_0, \dots, C_k] \rangle \langle D_0 \rangle.$$

Because of the reasoning from the previous proof, this is upper-bounded by $2\mathcal{L}_{\mathcal{I}}$. Notice that here we need to assume, as in the previous proof, that the Bell expression $\mathcal{I}[A_0, \dots, A_m, C_0, \dots, C_k]$ has no 1-body correlator terms for Charlie.

If Alice and Bob have a local deterministic strategy and Charlie and Dave share a NS resource, notice that $\langle D_1 \rangle = 0$, therefore, the second term vanishes. The first one becomes

$$\langle \mathcal{I}[A_0, \dots, A_m, C_0 D_0, \dots, C_k D_0] \rangle \langle B_0 + B_1 \rangle.$$

Now for all values of $\langle B_0 + B_1 \rangle \in \{0, \pm 2\}$ the $2\mathcal{L}_{\mathcal{I}}$ bound is satisfied. Here we need to assume that $\mathcal{I}[A_0, \dots, A_m, C_0, \dots, C_k]$ has no 1-body correlator terms for Alice.

Lastly, we consider the case where Alice and Bob share a NS resource and Charlie and Dave also share a NS resource. Notice that in this case, the 4-body correlator still factorizes between the two pairs because of the definition of broadcast nonlocality, $\langle A_i B_j C_k D_l \rangle = \langle A_i B_j \rangle \langle C_k D_l \rangle$. The second term of the inequality vanishes and the first one, because of the factorization, can be written as

$$\langle \mathcal{I}[A_0(B_0 + B_1), \dots, A_m(B_0 + B_1), C_0 D_0, \dots, C_k D_0] \rangle.$$

It is also clear from the arguments in the previous proof that this is upper bounded by $2\mathcal{L}_{\mathcal{I}}$.

# E  CO$_2$ emission table

Estimation for $CO_2$ emissions resulting from our numerical analysis, calculated using the examples of Scientific CO$_2$nduct [81]. Our emissions are equivalent to a car travelling 1350 km with an average emission rate of 0.122 $\text{kg}_{CO_2}$/km. The road distance from Barcelona to Vienna is 1782 km.

| Numerical simulations in Barcelona | |
|---|---|
| Total Kernel Hours [h] | ≥1440 |
| Thermal Design Power [W] | 165 |
| Total Energy Consumption Simulations [kWh] | 237.6 |
| Average Emission Of $CO_2$ In Spain [kg/kWh] | 0.265 |
| $CO_2$-Emission from Numerical Simulations [kg] | 163 |
| **Numerical simulations in Vienna** | |
| Total Kernel Hours [h] | ≥1200 |
| Thermal Design Power [W] | 15 |
| Total Energy Consumption Simulations [kWh] | 18 |
| Average Emission Of $CO_2$ In Vienna [kg/kWh] | 0.085 |
| $CO_2$-Emission from Numerical Simulations [kg] | 1.53 |
| Were The Emissions Offset? | **No** |
| Total $CO_2$-Emission [kg] | ≥164.64 |

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
