# Peer review of "Device-independent and semi-device-independent entanglement certification in broadcast Bell scenarios"

_SciPost Physics Core, doi:SciPost Phys. Core 6, 028 (2023)_

## Round 3 · Referee Report · Anonymous (Referee 1) · 2022-7-12

Strengths

  1. This paper provides arguably the strongest form of activation of Bell nonlocality with a single copy of a state. In particular, they show that a state which has a local-hidden-variable model in the 'standard' Bell scenario can in fact lead to genuine multipartite nonlocality in the broadcast scenario. This is a very nice result.
  2. They demonstrate that the broadcasting scenario is very powerful for device-independent entanglement certification, by showing that the two-qubit isotropic/Werner state can be activated in this scenario for (almost) the entire range when it is entangled. This is also an important result.
  3. They show that broadcast nonlocality can be generalized to broadcast steering (in a rather natural way), and show that this is able to activate steering. They show that this is able to activate the steering of the qubit isotropic/Werner state. This is the first time that the steering of this state has been activated using a single copy. This is also a nice result.

Weaknesses

  1. There is no intuition given for why the general method works or is a good method. This isn't necessary, but it may help in generating further work, as I assume that further results and methods should be sought which are able to promote standard Bell inequalities into broadcast Bell inequalities.
  2. There is no discussion of whether the general method for Bell inequalities also works for steering inequalities. This seems like a natural question to me (maybe I missed it somewhere, but I would expect a general result/section as in the nonlocality case?) 3.

Report

In my view, this paper certainly meets the journal's acceptance criteria. The important problem in the field that this paper addresses is the fundamental question of the relation between nonlocal effects and entanglement. It uses the novel method of the broadcast scenario (introduced recently in part by some of the authors) and significantly extends the results in this direction. It then extends this methodology beyond Bell nonlocality to EPR steering, and this constitutes an above-the-norm level of originality in my view. As outlined in the strengths above, the paper contains numerous significant results, and there is no question that this significantly advances the field in a number of ways. All 6 of the general acceptance criteria have also been met to my satisfaction. For this reason, my recommendation is to accept the paper, after addressing my minor comments below.

Requested changes

  1. (typo) "it is known that entanglement alone is not sufficient to observer neither Bell nonlocality nor EPR steering". The double negative is incorrect, and could cause confusion.
  2. (claim) You say that "the broadcast scenario requires the manipulation of a single copy of the state per experimental round and is therefore within the reach of available technology". This claim isn't obviously justified to me, due to the necessity of a broadcasting channel. Since this involves the preparation of ancillary systems being prepared (and potentially stored in quantum memories), and the controlled interactions before distribution, I see a number of realistic challenges in implementing this scenario. This isn't to criticize the scenario, however I feel that this statement is potentially not as justified as the text would make it appear.
  3. (typo) "lower detection inefficiencies" should be "lower detector efficiencies".
  4. (repetition) The paragraph at the top of page 6 to me felt like a rather large repetition of what was already (nicely) said in the introduction. Maybe the authors could consider harmonizing what is said, or removing detail from either place accordingly.
  5. (structure) Around (8) it might be useful to add a comment regarding that use of non-signalling behaviours, and to say that restricting further to quantum would lead to device-independent entanglement certification (see later section). Along this line, it might be worth stating more clearing in the introduction, for a non-expert, the distinction between device-independent entanglement certification and nonlocality, as I believe for many people the two may appear to be one and the same thing.
  6. (Add intuition). In Sec III A, it would be good to add some explanation or intuition for why (17) is a good construction. At the moment, it is only by analogy to (11), but this then doesn't explain why (11) captures broadcast nonlocality. Also, why is the restriction necessary? What role does having no local correlator play? Is there any hope for finding similar construction for more general Bell inequalities (not based upon correlators)?
  7. (concern). My one scientific concern regards the section on inefficiency. In the standard setting, we don't need to distinguish between the different sources of inefficiency; whether this comes from the transmission or the detectors, since all the inefficiency ends up multiplying, and we can just consider the total. Here, however, I believe the situation isn't so simple. If there is inefficiency in the transmission to Bob, this occurs before the broadcast channel, and so I believe it is confounded with the further transmission inefficiencies which send the state to Bob' and Charlie (or maybe just Charlie if we assume Bob'=Bob). On top of this we should then consider the detector efficiencies. It is therefore unclear to me if (21) is really 'fair' as a comparison distribution, as it may not fairly take into account of how the broadcasting scenario interacts with inefficiencies. I would stress that I don't think this is just being pedantic, but if I am correct, would really have implications for experimental implementations. I would therefore like to ask the authors to carefully consider whether this model is indeed the correct and fair model to study, and whether the results presented are mostly of theoretical interest, or capture the physics of losses.
  8. (presentation). I was not able to entirely follow the reasoning in the section "Broadcast activation without a broadcast channel". I would ask the authors to consider improving the presentation here, to make it clear precisely what is being claimed.
  9. (novelty). It isn't clear if the method presented in Appendix B is claimed to be novel, but this is already known. See for example (2) from https://doi.org/10.1103/PhysRevX.2.031003 or (B1) from https://doi.org/10.1103/PhysRevA.92.022354.

  • validity: top
  • significance: high
  • originality: high
  • clarity: high
  • formatting: excellent
  • grammar: excellent

Author:  Flavien Hirsch  on 2022-12-26  [id 3188]

(in reply to Report 1 on 2022-07-12)

We acknowledge the reviewer for the careful reading of our manuscript and various constructive comments. We are also happy to read that the report is favourable for publication.

For the upcoming resubmission, we have corrected all the typos and below we address all the technical comments raised by the reviewer.

Reviewer: 2. (claim) You say that "the broadcast scenario requires the manipulation of a single copy of the state per experimental round and is therefore within the reach of available technology". This claim isn't obviously justified to me, due to the necessity of a broadcasting channel. Since this involves the preparation of ancillary systems being prepared (and potentially stored in quantum memories), and the controlled interactions before distribution, I see a number of realistic challenges in implementing this scenario. This isn't to criticize the scenario, however I feel that this statement is potentially not as justified as the text would make it appear.

Authors: Indeed, we agree with the reviewer that the claim “therefore within the reach of available technology” is not obviously justified. We recall that the complete paragraph was:

“This has practical implications, since although stronger examples of activation are known by using many copies, the broadcast scenario requires the manipulation of a single copy of the state per experimental round and is therefore within the reach of available technology.”

There, our goal was simply to the contrast between the broadcast nonlocality scenario and the multicopy scenario, where two parties share several copies of a quantum states and perform joint measurements on such copies to activate nonlocality. However, we agree that this difference between the multicopy scenario and the broadcast scenario does not immediately imply feasibility with current technology.

We have now edited our previous phrase to:

“This has practical implications, since although stronger examples of activation are known by using many copies, the broadcast scenario requires the manipulation of a single copy of the state per experimental round, does not require joint measurements, and may admit a simpler implementation.”

Reviewer: 4. (repetition) The paragraph at the top of page 6 to me felt like a rather large repetition of what was already (nicely) said in the introduction. Maybe the authors could consider harmonizing what is said, or removing detail from either place accordingly.

Authors: We have edited that paragraph (on top of page 6) to make it less redundant with the introduction.

Reviewer: 6. (Add intuition). In Sec III A, it would be good to add some explanation or intuition for why (17) is a good construction. At the moment, it is only by analogy to (11), but this then doesn't explain why (11) captures broadcast nonlocality. Also, why is the restriction necessary? What role does having no local correlator play? Is there any hope for finding similar construction for more general Bell inequalities (not based upon correlators)?

Authors: We share the frustration of the reviewer about the lack of intuition regarding construction (17). We have put considerable effort in trying to arrive at (11) from first principles —without relying on numerical results from the linear program—to no avail. The restriction is necessary in order to prove the upper bound <I[A0,...,Am,C0,...,Ck](B0+B1)> <= 2L_I for the case of non-signalling extremal strategies for Bob and Charlie, as done in Appendix D. In the proof, the 1-body correlator terms of Alice lose their contribution to the inequality as they are multiplied by <B0+B1> which is zero. To extend our proof to inequalities with 1-body correlators, we need to find a way to express the local bound of the inequality without the 1 body correlators in terms of the local bound of the starting inequality, L_I, which we were unable to do. It is unclear if there is a physical significance to the lack of 1-body correlators (i.e., there is something to be understood about the broadcast scenario from this) or it is simply lack of technical knowledge. Regarding more general inequalities, there is no reason why similar constructions should not exist. In the literature, there is very little known about extremal non-signalling strategies involving more than two outcomes per party, which limits our ability to prove local bounds for inequalities with more than 2 outputs without using numerical techniques.

Reviewer: 7. (concern). My one scientific concern regards the section on inefficiency. In the standard setting, we don't need to distinguish between the different sources of inefficiency; whether this comes from the transmission or the detectors, since all the inefficiency ends up multiplying, and we can just consider the total. Here, however, I believe the situation isn't so simple. If there is inefficiency in the transmission to Bob, this occurs before the broadcast channel, and so I believe it is confounded with the further transmission inefficiencies which send the state to Bob' and Charlie (or maybe just Charlie if we assume Bob'=Bob). On top of this, we should then consider the detector efficiencies. It is therefore unclear to me if (21) is really 'fair' as a comparison distribution, as it may not fairly take into account of how the broadcasting scenario interacts with inefficiencies. I would stress that I don't think this is just being pedantic, but if I am correct, would really have implications for experimental implementations. I would therefore like to ask the authors to carefully consider whether this model is indeed the correct and fair model to study, and whether the results presented are mostly of theoretical interest, or capture the physics of losses.

Authors: We mostly agree with the referee on that comment. Depending on the experimental implementation of the broadcast scenario, one will not only have inefficiencies coming from detector imperfections (losses, etc.), but also from transmission of the state, and in the case of the broadcast scenario, potential inefficiencies coming from the implementation of the channel are expected.

The rationale behind our analysis is that previous studies of detector inefficiencies focus on the noise from the detectors and neglect the other sources of noise. Therefore, we adopt the same approach so that we can meaningfully benchmark our results against the existing literature, e.g., the https://doi.org/10.1103/PhysRevA.47.R747, https://doi.org/10.1103/PhysRevA.88.014102, and https://doi.org/10.1103/PhysRevLett.98.220403;

To make that clearer, we now add a paragraph clarifying this point and mentioning the potential other sources of experimental noise which should appear in a realistic scenario.

Reviewer: 8. (presentation). I was not able to entirely follow the reasoning in the section "Broadcast activation without a broadcast channel". I would ask the authors to consider improving the presentation here, to make it clear precisely what is being claimed.

Authors: We rewrote that paragraph, making the reasoning and the claim clearer.

Reviewer: 9. (novelty). It isn't clear if the method presented in Appendix B is claimed to be novel, but this is already known. See for example (2) from https://doi.org/10.1103/PhysRevX.2.031003 or (B1) from
https://doi.org/10.1103/PhysRevA.92.022354.

Authors: We were unaware that the optimisation method for steering inequalities written in Appendix B was already known and presented in these references. We acknowledge the reviewer for pointing this out. In our new version of the manuscript, we now do not claim the result as new, and we include a proper citation for these two references.

---

## Editorial Decision

published